# Unravelling the Potential of *Lactococcus lactis* Strains to Be Used in Cheesemaking Production as Biocontrol Agents

**DOI:** 10.3390/foods9121815

**Published:** 2020-12-07

**Authors:** Danka Bukvicki, Lorenzo Siroli, Margherita D’Alessandro, Sofia Cosentino, Ismail Fliss, Laila Ben Said, Hebatoallah Hassan, Rosalba Lanciotti, Francesca Patrignani

**Affiliations:** 1Department of Morphology and Systematics of Plants, Faculty of Biology, Institute of Botany and Botanical Garden “Jevremovac”, University of Belgrade, Takovska 43, 11000 Belgrade, Serbia; dankabukvicki@gmail.com; 2Department of Agricultural and Food Sciences, University of Bologna, p.zza Goidanich 60, 47521 Cesena, Italy; lorenzo.siroli2@unibo.it (L.S.); margheri.dalessandr3@unibo.it (M.D.); rosalba.lanciotti@unibo.it (R.L.); 3Interdepartmental Center for Industrial Agri-Food Research, University of Bologna, Piazza Goidanich 60, 47521 Cesena (FC), Italy; 4Department of Medical Sciences and Public Health, University of Cagliari, 09042 Monserrato, Italy; scosenti@unica.it (S.C.); hebatoallah.hassan.1@ulaval.ca (H.H.); 5Food Science Department, Food and Agriculture Faculty, Institute of Nutrition and Functional Foods, Laval University, 2425 Agriculture Street, Quebec City, QC G1V0A6, Canada; ismail.fliss@fsaa.ulaval.ca (I.F.); laila.ben-said.1@ulaval.ca (L.B.S.)

**Keywords:** *Lactococcus lactis*, milk, nisin, volatile compounds, *Listeria monocytogenes*, antimicrobial activity

## Abstract

This research, developed within an exchange program between Italy and Canada, represents the first step of a three-year project intended to evaluate the potential of nisin-producing *Lactococcus lactis* strains isolated from Italian and Canadian dairy products to select a consortium of strains to be used as biocontrol agents in Crescenza and Cheddar cheese production. In this framework, the acidification and the production of nisin in milk, and the volatile molecule profiles of the fermented milk, were recorded. The strains were further tested for their anti-*Listeria monocytogenes* activity in milk. The data obtained highlighted good potential for some of the tested strains, which showed production of nisin beginning within 12 h after the inoculation and reaching maximum levels between 24 and 48 h. The highest inactivation levels of *L. monocytogenes* in milk was reached in the presence of the strains 101877/1, LBG2, 9FS16, 11FS16, 3LC39, FBG1P, UL36, UL720, UL35. The strains generated in milk-specific volatile profiles and differences in the presence of fundamental aromatic molecules of dairy products, such as 2-butanone and diacetyl. The results highlight the interesting potential of some *L. lactis* strains, the producer of nisin, to be further used as biocontrol agents, although the strains need to be tested for interaction with traditional thermophilic starters and tested in real cheesemaking conditions.

## 1. Introduction

Cheeses and dairy products are widely consumed, and they represent one of the fastest growing sectors within the food industry [1,2]. In specific cases, some types of cheese are susceptible to microbial spoilage and they can also be carriers of foodborne pathogenic species deriving from raw milk. In fact, sometimes, these microorganisms can be resistant to the pasteurization process of the milk and survive in the final product. Many attempts have been made to reduce the incidence of pathogenic and spoilage agents, including using chemical additives, which are usually perceived negatively by consumers [3]. To solve this challenge, in the past few decades, industrial and academic research has focused its attention on natural antimicrobials since the expectations of consumers toward natural products have increased significantly. In the dairy industry, this outcome can be achieved by the use of natural antimicrobial compounds produced in situ by lactic acid bacteria (LAB) [4]. In fact, LABs can produce many antimicrobial molecules such as organic acids, diacetyl, hydrogen peroxide, peptides with antifungal action and bacteriocins [5,6]. Bacteriocins, short peptides with a low molecular weight, are considered safe food biopreservatives because they are simply degraded in the human gastrointestinal tract and do not negatively impact the gut microbiota [7,8,9,10]. Production of bacteriocins by LAB could be important to manage and reduce the growth of food pathogens and spoilage bacteria in food [4]. Nevertheless, some limiting factors such as absorption of food components, enzymatic degradation, and poor solubility need to be taken into consideration [10]. Among LAB, *Lactococcus lactis* is one of the most frequently researched species due to its aptitude to produce nisin, a bacteriocin largely studied in different foods for its wide antimicrobial spectrum [11]. In particular, nisin is especially effective against Gram-positive bacteria such as *Listeria monocytogenes*, *Clostridium* spp., and *Staphyloccoccus aureus* [12,13,14,15]. The effectiveness of nisin has been demonstrated in vitro and in situ [12,14] including vegetable-based juices and beverages [13,15], fermented vegetable products [16], as well as meat-based foods [17]. This bacteriocin, and particularly the variant Z, is very soluble and stable at high temperatures and in acidic environments [10]. Although its antimicrobial activity has been well tested, the European Food Safety Authority (EFSA) only allows its use as a biopreservative (E234) in a few foods (cheeses like mascarpone and some egg-based preparations). Due to several strict regulations, the dairy industry is interested in searching for nisin-producing *L. lactis* strains to be used as a biopreservation strategy as a part of the fermentation process or as adjuvant cultures [10]. For the dairy industry, this represents a great advantage since *L. lactis* is widely used as starter, and/or co-starter for several cheese productions, including Cheddar and Crescenza [18,19]. Moreover, the incorporation of nisin-producing *L. lactis* strains could also be a strategy to increase the desirable sensory features of the final dairy products. However, for this purpose, the selection of suitable *L. lactis* strains needs to take into account some important safety and technological considerations, such as the strain’s antibiotic resistance [20,21], the strain’s ability to quickly ferment the substrate and produce acid (in case it is used as a starter), to impart good sensory properties, and to inhibit the growth of pathogenic microorganisms (especially if used as biocontrol agents) [10,21].

In this framework, the main objective of the present study, developed by an exchange program between Italy and Quebec, was to characterize selected *L. lactis* strains, isolated from Italian and Canadian dairy products in order to use some of them as potential protective cultures in cheesemaking to produce Cheddar (hard cheese) or Crescenza (soft cheese). Twenty strains from two geographical origins (Italy and Canada) were screened for (some) important technological traits and their ability to generate volatile molecule profiles in milk at two different incubation temperatures. Additionally, in order to select good biocontrol agents, the nisin production over the incubation and the ability to inhibit *Listeria monocytogenes* in skim milk were also screened for all the strains.

## 2. Materials and Methods 

### 2.1. Bacterial Strains and Culture Media

The strains employed in this research belong to the collection of the Department of Agricultural and Food Sciences (University of Bologna), the Department of Public Health, Clinical and Molecular Medicine (University of Cagliari) and STELA research center (Universite Laval, Quebec, QC, Canada) as reported by Table 1. The strains were isolated from different dairy products and were already identified as having the gene-encoding nisin. All the strains were maintained at −80 °C in M17 medium (Oxoid, Milan, Italy) supplemented with 30% glycerol prior to their use (Sigma Aldrich, Milan, Italy). They were also grown twice in M17 medium (Oxoid, Milan, Italy) for 24 h at 30 °C under aerobic conditions before being used.

### 2.2. Detection of Nisin Encoding Gene

For the strains belonging to the University of Bologna collection, total genomic DNA was extracted from *L. lactis* cells and purified using the InstaGene matrix mix (Bio-Rad laboratories, Milano, Italy), following the manufacturer’s recommendations. In the analysis, a negative control and strains encoding for nisin Z (*L. lactis* CBM21, [22]) and nisin A (*L. lactis* ATCC11454, [23]) as positive controls were used. The amplification of the nisin-encoding gene was performed by using the primers reported by De Vos et al. [24] (forward: 5’-CGCGAGCATAATAAACGGCT-3’; reverse: 5’-GGATAGTATCCATGTCTGAAC-3’). The composition of the PCR mixture (50 µL) was the following: 2 mM MgCl_2_, 0.2 µM of each primer, 0.2 mM of deoxyribonucleotide triphosphates (dNTPs), 0.02 U/µL Taq polymerase, 1× PCR buffer, and approximately 20 ng of genomic DNA [22]. The thermocycling conditions were the following: denaturation at 94 °C for 5 min; 30 cycles at 93 °C for 2 min, 54 °C for 1 min, 72 °C for 1.5 min, then a final extension at 72 °C for 10 min. PCR products were visualized on 1.5% agarose gel. The PCR resulting amplicons were purified with the QIAquick PCR Purification Kit (Qiagen, USA) and sequenced at BMR Genomics sequencing centre (Padua, Italy). Sequences were then compared with those available in GenBank database retrieved through BLASTn searches and then aligned using the GeneDoc 2.7 software as reported by Siroli et al. [22].

### 2.3. Fermentation Ability and Acidification in Milk

The strains were sub-cultured twice in M17 medium at 30 °C for 24 h and further inoculated in 100 mL of commercial UHT whole cow milk at a final concentration of approximately 6 log cfu/mL in triplicate. The samples were incubated at 30 and 37 °C. Immediately after inoculation and after 12, 24, 48, and 72 h, the pH decrease was monitored using 8519 Hanna-Instruments pH-meter (Milan, Italy) and *L. lactis* cell counts were detected. For this, decimal dilutions were performed in distilled water added of 0.9% NaCl and spread onto M17 agar plates added with 0.5% lactose. The Petri dishes were incubated at 30 °C for 24–48 h in aerobic conditions.

### 2.4. Volatile Molecule Profiles of Fermented Samples

The analysis of volatile molecule profiles was performed by gas chromatography–mass spectrometry (GC/MS) coupled with solid phase micro extraction (SPME). After 72 h of incubation, milk samples, inoculated with different strains of *L. lactis,* were analysed according to the method of Patrignani et al. [25]. Briefly, 5 millilitres of sample were put in vials and incubated for 10 min at 45 °C, then the fibre (CAR/PDMS, 75 μm, SUPELCO, Bellafonte, PA, USA) was left into the vial head space for 30 min at 45 °C. The absorbed volatiles were desorbed in the gas chromatograph (GC) injector port in spitless mode at 250 °C for 10 min. Headspace of the volatile compounds was analysed using Gas-Chromatography (GC) 6890N, Network GC System with mass spectrometry (MS) 5970 MSD (Hewlett–Packard, Geneva, Switzerland). The Chrompack CP-Wax 52 CB (50 m × 320 μm × 1.2 μm) column was used. The initial temperature was 40 °C for 1 min, and then increased by 4.5 °C/min up to 65 °C. Then, temperature increased by 10 °C/min up to 230 °C and remained at this temperature for 17 min. Gas-carrier was helium at 1.0 mL/min flow. The different molecules were recognised by comparison based on NIST 11 (National Institute of Standards and Technology) database. Moreover, an internal standard (4-methyl-2-pentanol) was used.

### 2.5. Nisin Activity Determination

The activity of nisin was detected in inoculated commercial skim milk samples during the fermentation at 30 and 37 °C, after 6, 12, 24, 48, and 72 h of incubation, in relation to the strain used. Nisin assay was performed by the agar-well diffusion method as described by Pongtharangkul and Demirci [26], Millette et al. [27], and de Oliveira Junior et al. [28] with some modifications. Ten milliliters of the fermented sample were collected, and pH of the samples was adjusted to 6 with 2 N NaOH and then centrifuged at 6000 g for 20 min. The supernatants were collected and filtered (0.45 μm pore diameter). Next, supernatants were let to boil for 10 min and then chilled. Wells of 5 mm diameter were made into each plate and filled with 50 μL of the supernatant. *Lactobacillus plantarum* V7B3 (collection at Department of Agricultural and Food Sciences, University of Bologna, Italy) was used as the nisin-sensitive strain in the study [6]. The strain was grown in de Man, Rogosa, and Sharpe (MRS, Oxoid, Milano, Italy), respectively, for 24 h at 37 °C. The agar-well diffusion assay was performed in MRS soft agar (0.8% agar) inoculated with *L. plantarum* V7B3 at 7.0 log CFU/mL. The zones of inhibition were measured after incubation at 37 °C for 24 h and related with standard nisin linear regression equation to obtain final concentrations.

### 2.6. Standard Nisin Linear Regression Equation

A stock solution of nisin was prepared by dissolving commercial nisin 2.5% (Sigma-Aldrich, Milan, Italy) in 0.02N HCl, which had been previously sterilized. Concentrations ranging from 1000 to 0 International Units (IU) (1000, 500, 400, 300, 200, 100, 50, and 0 IU/mL) were prepared in skim milk and the nisin activity was plotted against these concentrations to construct the standard curve according to the method previously described using the same substrate and temperature condition. The line regression equation was determined the for standard curve. The activity of nisin expressed in international units per milliliter was converted to mg/L through the relation in Equation (1):nisin (mg/L) = (z × 0.025),(1)
where z = IU/mL and 0.025 is a conversion value related to 2.5% pure nisin.

### 2.7. Challenge Test in Presence of Listeria Monocytogenes

The strains were tested to evaluate their ability to inhibit *L. monocytogenes* in milk at 30 and 37 °C. *Lactococcus* strains were inoculated at 6.0 log cfu/mL in commercial UHT whole milk, while *L. monocytogenes* (Scott A) was inoculated at 4.0 log cfu/mL. The samples were performed in triplicates. *L. lactis* counts were detected on M17 agar supplemented with a solution of lactose 10% (Oxoid, Milan, Italy) while Listeria selective agar base added of Listeria selective supplement (Oxoid) was used for *L. monocytogenes* counting. Samples were taken at 0, 6, 12, 24, 48, and 72 h of incubation at 30 and 37 °C.

### 2.8. Statistical Analysis

The data are expressed as the mean of three repetitions and two independent experiments. The cell load and pH data were considered significantly different (*p* < 0.05) on the basis of ANOVA and TUKEY HSD post hoc. The volatile molecule raw data (expressed as area) were analyzed using ANOVA followed by a principal component analysis (PCA), performed by Statistica software (Stat 5.0 for Windows), to define the effects of the individual strains and the temperature of incubation on the volatile compounds produced.

## 3. Results

### 3.1. Detection of Nisin-Encoding Gene

In order to identify the type of produced nisin for LSGA1B, FBG1P, LBG1G, LSG3, LBG2), a 598-bp fragment was amplified from the genomic DNA of these strains, which was identical to that of nisin-positive control strains *L. lactis* subsp. *lactis* ATCC 11454 and *L. lactis* subsp. *lactis* CBM21 (Figure not showed).

The sequencing of amplicons resulting from PCR are reported in Figure 1. The sequences, compared with those available in GenBank database retrieved through BLASTn search, confirmed the production of Nisin A for LSGA1B, FBG1P, LBG1G and nisin Z for LSG3, LBG2.

### 3.2. Lactococcus lactis Growth and Ph Decrease in Whole Milk at 30 and 37 °C

The growth and the pH decrease of *L. lactis* strains in whole commercial UHT cow milk at 37 °C are put in Figure 2 and Figure 3, respectively. All the strains inoculated in milk at a level between 6.0 and 6.5 log cfu/mL were able to reach the maximum cell loads during 12 h at 37 °C, achieving over 8.0 log cfu/mL. Only the strain 9FS16 reached cell load values over 8.0 log cfu/mL between 48 and 72 h. However, after 72 h of incubation this strain’s growth was not significantly different from that of most of the tested strains. After 72 h of incubation at 37 °C, among Canadian strains, significant cell load decreases were reached by the strains UL719 and 730. On the other hand, these two strains showed a similar behaviour when incubated at 30 °C (Figure 4). As reported in Figure 3, most of the strains were able to reduce the pH of the milk to levels below 5.0 over 72 h of incubation at 37 °C. In general, the strains belonging to the collection of STELA were faster with respect to the other strains in the first 12 h of incubation since they were able to reduce the pH of milk at levels ranging between 5.51 and 5.79.

In Figure 4 and Figure 5, the cell load and the pH decrease of the *L. lactis* strains inoculated in cow milk and stored at 30 °C are reported over time. All the strains reached cell load levels higher than 8.0 log cfu/mL within 24 h except for the strain 9FS16, which had significantly lower levels when compared with the remaining strains. Additionally, in this case, some of the strains from the Canadian collection (UL 36, UL35, UL719, UL720) showed rapid acidification and growth, reaching values of 8 log cfu/mL in 12 h. Of note, the strains 6/23898, FBG1P, and LBG2 were able to significantly decrease the pH of milk below 5.0 in 24 h.

### 3.3. Volatile Aroma Profile by GC-MS/SPME at 30 and 37 °C

The volatile molecule profiles, detected in milk samples by GC–MS/SPME after 72 h of incubation at 30 and 37 °C, showed different aromagram profiles according to the strain and incubation temperature used. The main chemical classes of compounds were represented by ketones, alcohols, acids, and aldehydes. To highlight the differences in the volatile compound profiles of fermented milks at 37 and 30 °C in relation to each individual strain, the GC–MS/SPME raw data (expressed as area) were processed by PCA. Data from samples incubated at 37 °C are reported in Figure 6a,b, while Figure 7a,b shows data from samples incubated at 30 °C. After fermentation at 37 °C, the samples were projected on the factorial plane (1 × 2) spanned by the first two factors (Factor 1 and Factor 2), which describe 46.83 and 14.1%, respectively, of the total variance among the samples. The separation on the factorial plane was mainly related to the biovar. In fact, strains with Canadian origins were grouped together with the exception of the strain UL 33. The molecules that affected this behavior (as reported in Figure 6b) were mainly 2,3 butanedione, 3-hydroxy2-butanone, 1-heptanone, 2 methyl propanol, and acetic acid. Another group, separated from the previous one along Factor 2, was clearly composed by the metabolites coming from the strains 1LC18, FBG1P, 9FS16, 3LC39, 6/23898, 11FS16, 10/18771, UL33. In this case, 2-butanone, 2-ethyl, decanol, acetone, ethanol, 2-pentanone, 2-Heptanol, 6 methyl, 3-methyl butanal, methyl isobutyl ketone contributed to the grouping. The third group, separated from the others, along Factor 1, was represented by the remaining samples whose clusterization was affected by 1 hydroxy-2 propanone, butanoic acid, decanoic acid, benzenacetaldehyde, 2,6 dimethyl 4 heptanol, 3 methyl decanoic acid.

Figure 7a,b show, respectively, the loading plots of strain cultures in milk at 30 °C and the volatile compounds produced in the plane defined by principal components 1 (Factor 1) and 2 (Factor 2). In this case, Factor 1 described the 45.85% variance among samples while Factor 2 accounted for 17.14%. Additionally, in this case, samples obtained by STELA collection, with the exception of UL33, grouped together, including samples from 9/20234 and 9FS16. This group, separated from other strains along Factor 1 and Factor 2, was affected by 2,3-butanedione, 3-hydroxy-2-butanone, 2,6 dimethyl heptanone, 4-methyl isobutyl ketone, 5-ethyl-3-nonanol, 2-methyl-3-pentanol, 3-penten-2-one-4-methyl. Samples from LBG2, 101877/1, and FBG1P tended to group together and the compounds affecting this were phenylethyl alcohol, acetone, benzaldehyde, benzenacetaldehyde, 3-methyl butanal, 2,6-dimethyl heptanol. The grouping of the remaining samples (UL33, 6MRSL55, 1LC18, 3LC39, 11FS16, LBG1G, and LBG3) was affected by ethanol, 2-methyl-1-propanol, acetaldehyde, hexanoic acid, and butanoic acid.

### 3.4. Nisin Production in Skim Milk at 30 and 37 °C

The nisin production by the tested strains in skimmed milk, after 12, 24, 48, 72 h of incubation at 30 and 37 °C, are shown in Figure 8a,b. In general, *Lactococci* strains produced higher concentration of nisin, expressed as ppm, at 30 °C rather than 37 °C. According to the data shown in Figure 8a, the Italian strains able to produce the highest nisin values at 30 °C were 11FS16, 101877/1, and LBG2. In particular, these strains produced, respectively, 16 ppm at 48 h, 13 ppm at 48 h, and 13.3 ppm at 24 h. The strain 11FS16 was able to produce at 37 °C amounts of nisin comparable to those produced at 30 °C at 48 h. Regarding STELA strains, the strain UL33 (after 24 h of incubation at 30 °C) produced about 10 ppm of nisin, while at 37 °C, analogous production of nisin was delayed up to 48 h. Some of the tested strains (9FS16, FBG1P, 3LC39, and UL33) were able to produce nisin during the incubation at 30 °C within 6 h (data not shown). The largest production of nisin, among STELA strains, was reached by the strain UL 209 in 48 h at 30 °C. After 72 h of incubation, the concentration of nisin decreased in all the samples.

### 3.5. Interaction with Listeria Monocytogenes in Skim Milk at 30 and 37 °C

The interaction between the tested strains and *L. monocytogenes* Scott A is reported in Table 2. For each strain tested, inoculated in milk at 6.0 log cfu/mL, *L. monocytogenes* was inoculated at the level 4.0 log cfu/mL and the viable counts monitored during milk incubation at 30 and 37 °C.

After 24 h of incubation, the highest reductions in *L. monocytogenes* were found in the samples inoculated at 30 °C, regardless of the strain used. This data can also be explained by the *L. lactis* nisin production, which was higher at 30 °C than at 37 °C. According to the data for the strains 101877/1, 6MRSL55, LSG3, LBG2, 9FS16, FBG1P, 3LC39, UL35, UL36, UL719, UL209, UL720, the inactivation of *L. monocytogenes* was increased during storage, independently from the incubation temperature but in a strain-dependent way. For the strains LSGA1B, LBG1G, 11FS16, 6/23898, the maximum inactivation was reached between 24 and 48 h, suggesting that some strains have bacteriostatic effects.

## 4. Discussion

The present research, developed by an exchange project between Italy and Canada–Quebec, focused on the characterization of *L. lactis* strains in terms of their fermentation capabilities in cow milk, ability to generate good volatile profiles, the nisin production, and the ability to inhibit *L. monocytogenes* when co-inoculated in milk to select a consortium of strains to be further used for biopreservation in cheese production. The species *L. lactis* is one of the most import in the dairy industry and it is “generally recognized as safe” (GRAS) [29]. On the other hand, the EU has created a list of microorganisms with a long history of safe use based on a qualified presumption of safety (QPS) in which *L. lactis* strains are included [30]. In the present study, we focus on this species because it is normally used as starter for Cheddar cheese at 30 °C and as co-starter for Crescenza cheese at 37 °C. For this reason, the final aim of the exchange project will be the selection of a consortium including *L. lactis* nisin-producing strains to use as preservation cultures in these dairy productions. The strains were screened for their growth in milk at both the temperatures used in normal cheesemaking protocols. In the present research, all of the strains tested showed good ability to grow in whole cow milk and to decrease the pH at values below 5 within 72 h, independently of the temperature used. This trait could be very important in the case of Crescenza since soft cheese is more affected by bacterial spoilage with respect to ripened ones. Exceptions to this behavior were represented by the strains LBG1G, FBG1P, and ILC18, which decreased the milk pH up to 5.42, 5.67, and 5.50, respectively. On the other hand, the considered strains have mesophilic properties, having an optimum growth ranging between 20 and 35 °C. and, thus, most performing at 30 °C with respect to 37 °C. However, the good potential of these strains’ growth is very important since nisin production starts in the early exponential phase of the strains and it is maximized when strains reach the early stationary phase [31]. However, according to the findings of Chaves de Lima et al. [32], viable *L. lactis* cell loads show that optimal cell growth does not always result in a high bacteriocin titer, as previously reported in studies with this LAB species [33,34,35]. In fact, the optimum growth also depends on the features of the microorganism used and the environmental settings [36]. The best production of bacteriocins for technological applications in dairy sector remains a challenge.

According to the data obtained by GC/MS/SPME, the strains, inoculated in milk at 30 and 37 °C, generated specific aromagrams. The strains from the STELA collection were able to produce the highest amounts of 2,3-butanedione (diacetyl) and 2-butanone-3-hydroxy (acetoin) in milk when compared with the strains belonging to the University of Bologna and Cagliari. These molecules, with acetaldehyde and 2,3-butanediol, are usually considered as the key aroma of yoghurt- and soft-cheese-like products [37,38] and they can be considered acceptable for products like Crescenza. Particularly, diacetyl, deriving from the citrate metabolism, is an important key molecule of many dairy products since low concentrations impart a creamy and buttery aroma. It also represents a key compound of Camembert, Cheddar, and Emmenthal [39] and soft cheeses [40]. The ability of *L*. *lactis* to produce these key aromas is well known, and this is one of the main reasons why it is used at an industrial level [41,42]. In addition, the strains belonging to the Italian collection were able to produce acceptable volatility profiles characterized by ketones and alcohols but lower quantities of diacetyl.

However, it is important to emphasize that the final flavor and aroma of a food product is the result of several interactions between several chemical molecules and sensory receptors, which are affected, in turn, by the food matrix composition and microstructure [25,43].

To select appropriate strains to be used for biopreservation, the production of nisin, for each strain considered, was investigated in skimmed milk. The strains were able to produce the highest amount of nisin at 30 °C with respect to 37 °C. These data are in agreement with the reference data. In fact, Barman et al. [31] showed that for three strains of *L. lactis* isolated from buttermilk, the maximum extracellular bacteriocin production took place in MRS medium at 28 °C with respect to 37 °C. Additionally, Guerra and Castro [44] found that the maximum bacteriocin production for *L. lactis* subsp. *lactis* CECT 539, using mussel-processing wastes, was higher at 30 °C with respect to 37 °C. Additionally, Chaves de Lima et al. [32], by using a central composite design to optimize the production of bacteriocin-like substances from *L. lactis*, found optimal production at 28 °C using goat cheese, and at 31 °C when a probiotic substance was added to the substrate. The challenge test was performed in this research using *L. monocytoge*nes Scott A as a microbial target, since this strain is one of the most studied and applied in vitro studies. Although an additional strain, also from a food source, would need to be tested, contamination with *L. monocytogenes* is a major alarm in fermented foods. Numerous listeriosis outbreaks have been related to the eating of dairy products, particularly soft cheeses, causing problems for the dairy industry and public health authorities [45]. Despite the fact that most dairy products, especially cheese, are made from pasteurized milk, listeriosis still happens. Moreover, cheese products are ready to eat and are usually stored at refrigeration temperatures, permitting the survival and the growth of psychotropic pathogens. The data regarding the challenge test demonstrated that the highest inactivation of *L. monocytogenes* Scott A was reached in the presence of the strains 101877/1, LBG2, 9FS16, 11FS16, 3LC39, UL36, UL720, UL35, FBG1P. On the other hand, some of these strains were also able to produce the highest amounts of nisin, especially at 30 °C. Some of these strains produce nisin A, which is reported to be less effective with respect to nisin Z [10]. However, the inactivation of *L. monocytogenes* that was observed can be the result of several factors, including not only the type of nisin produced, but also the amount and the interaction with the system considered. Moreover, other factors, such as the ability of lactic acid bacteria to produce other antimicrobial substances such as lactic acid, hydrogen peroxide, or the competition for the substrate, can explain the highest decrease in *L. monocytogenes* cell loads observed due to the nisin-Z-producing strains [22]. Additionally, the fast acidification can be a useful tool to inactivate several pathogenic species such as *L. monocytogenes* [6,46]. In this research, the release of nisin production and *L. monocytogenes* inactivation was performed in milk in order to simulate a real condition as much as possible. In fact, the efficacy of bacteriocins in culture media is not always reproducible in food systems (in situ). In fact, numerous food variables, such as interaction with additives/ingredients, absorption of food components, and inactivation by food enzymes and pH changes in food, can affect the nisin inhibitory effect and its stability. For example, low solubility and uneven distribution in the food matrix and limited stability of bacteriocin during food shelf life are additional factors that influence the activity of bacteriocins in foods and, consequently, their antimicrobial activity against pathogenic microorganisms.

Additionally, the interaction between bacteriocin/food microbiota can be responsible for changes in the sensitivity to the bacteriocins. Similarly, the physiological state of the microbial target (growing, resting, starving, or viable but non-culturable cells, stressed, or sub-lethally injured cells, endospores), the protection by physical-chemical barriers (microcolonies, biofilms, slime), and the development of resistance/adaptation [47] play an important role in bacteriocins activity. Moreover, even if bacteriocin/s are produced in a sterile milk medium, antagonistic effects against *L. monocytogenes* are frequently influenced by pH, presence of NaCl, temperature, and other ingredients of the fresh or mature cheese. These factors have an impact on the interaction and absorption of bacteriocin/s to *L. monocytogenes* [48]. In the present research, most of the samples that were considered showed a marked decrease in the amount of nisin detected over time, independent of the strain considered. This behavior confirms the reference data highlighting that most of nisin production happens in the late exponential growth phase and the beginning of the stationary phase. Afterward, a reduction of the metabolic process leading to the production of the bacteriocin takes place since its biosynthesis is repressed by the bacteriocin accumulation in the growth media [49]. Furthermore, the chemical-physical and compositional features of the substrate are especially able of modifying the stability and activity of the bacteriocin over time. The bacteriocin can interact with other macromolecules present in the substrate, losing its antimicrobial activity [10,50]. Furthermore, nisin is disposed to reduction due to its physical diffusion within the food system or its degradation by proteases [51]. These reasons can explain why some strains such as LSGA1B, 9/20234, LBG1G, and 11 FS16 demonstrated more inactivation against *L. monocytogenes* at 48 h of incubation rather than 72 h.

## 5. Conclusions

The results that were generated point to the challenge of using some *L. lactis* strains as bio-control agents in milk. The strains generated specific volatile profiles that show clear differences in fundamental aromatic substances in dairy products such as 2-butanone and diacetyl. In addition, some strains showed that the rapid nisin production started within 12 h of fermentation, reaching maximum levels between 24 or 48 h, also inhibited and/or reduced the growth of *L. monocytogenes* Scott A in milk systems. Although other several strains of *L. monocytogenes*, also from food sources, would need to be tested, this work permitted to select the most promising nisin-producing strains to further use. However, the interaction of *L. lactis* strains with starters should be carefully studied in future research, especially for producing a cheese like Crescenza, which is created in the presence of thermophilic starters. Strains 3LC39, 101877/1 (producing high level of nisin), LBG2 (producing high level of nisin), 9FS16, UL35, UL36, and UL720 are good candidates as potential bio-control agents for *L. monocytogenes* in dairy products. However, before being used in cheese production, they need to be tested for a variety of interactions, including with starters, in real conditions.

## Figures and Tables

**Figure 1 foods-09-01815-f001:**
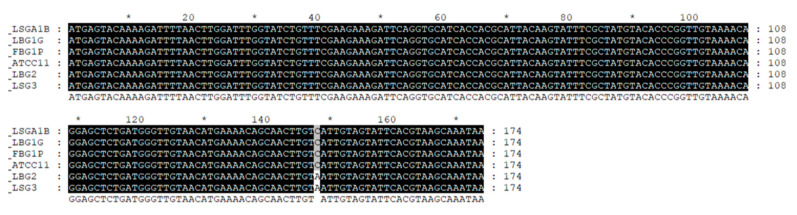
Multiple alignment of amplicon sequences resulting from PCR analysis for the strain of *L. lactis* FBG1P, LBG1G, LSG3, LBG2, LSGA1B.

**Figure 2 foods-09-01815-f002:**
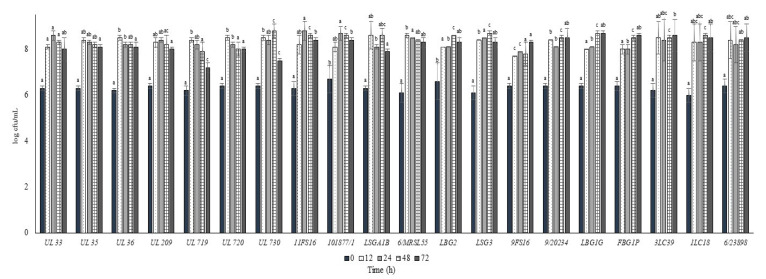
Cell loads (log cfu/mL) of the tested *Lactococcus lactis* strains in milk over the incubation at 37 °C. For each sampling time considered, samples with a different lowercase letter are significantly different (*p* < 0.05).

**Figure 3 foods-09-01815-f003:**
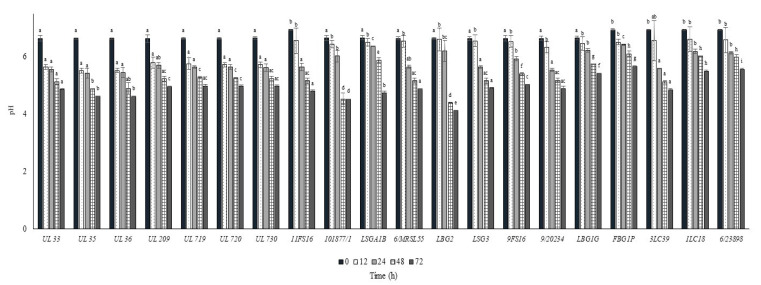
pH values ± SD recorded in milk samples after the inoculum of the *Lactococcus lactis* strains and the incubation at 37 °C. For each sampling time considered, samples with a different lowercase letter are significantly different (*p* < 0.05).

**Figure 4 foods-09-01815-f004:**
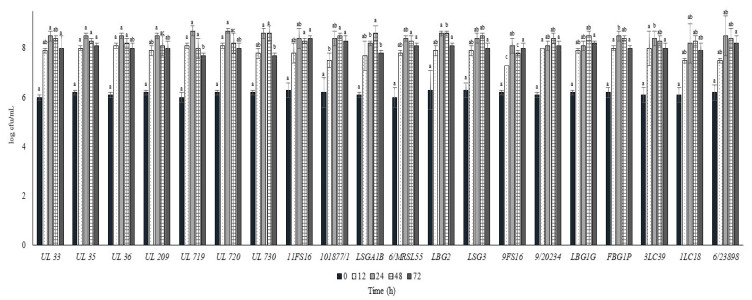
Cell loads (log cfu/mL) of the tested *Lactococcus lactis* strains in milk over the incubation at 30 °C. For each sampling time considered, samples with a different lowercase letter are significantly different (*p* < 0.05).

**Figure 5 foods-09-01815-f005:**
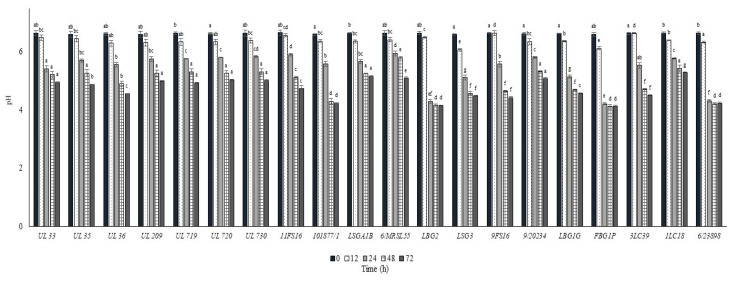
pH values ± SD recorded in milk samples after the inoculum of the *L. lactis* strains and the incubation at 30 °C. For each sampling time considered, samples with a different lowercase letter are significantly different (*p* < 0.05).

**Figure 6 foods-09-01815-f006:**
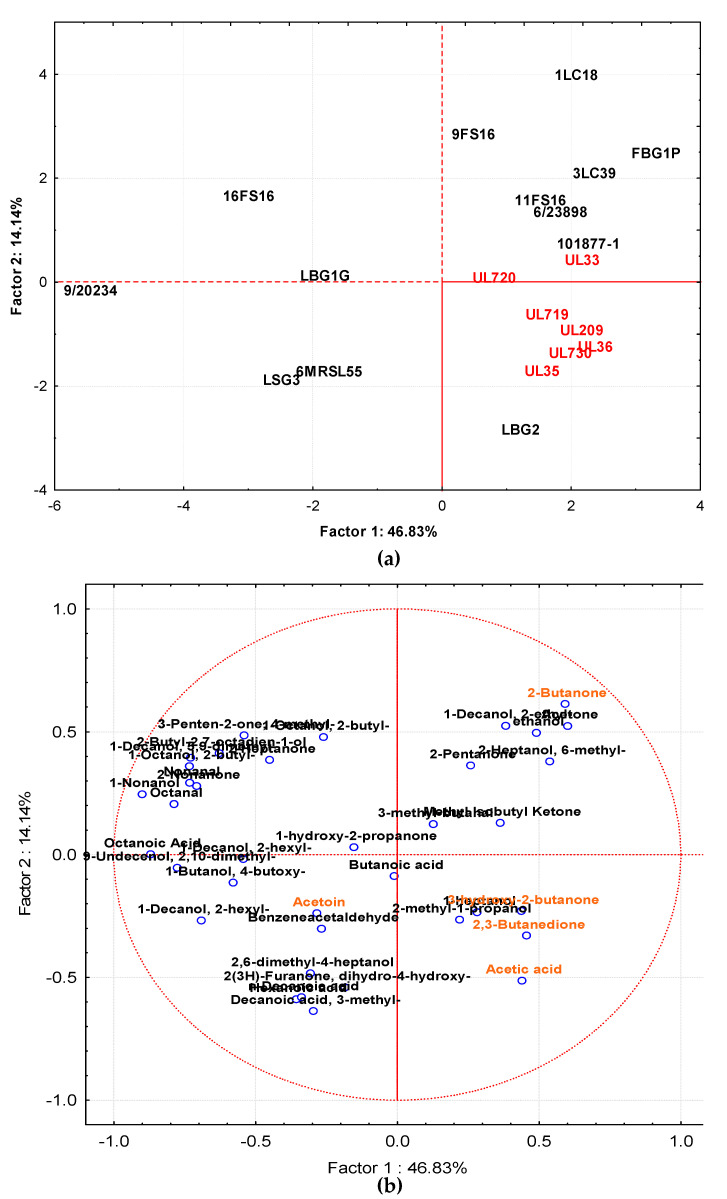
Distribution of cases (**a**) and variables (**b**) on the factor plane (1 × 2) obtained by samples fermented at 37 °C. Factor 1 and Factor 2 explain 46.83% and 14.14% of the total variance, respectively. The sample obtained by strains isolated from Canadian products are highlighted in red.

**Figure 7 foods-09-01815-f007:**
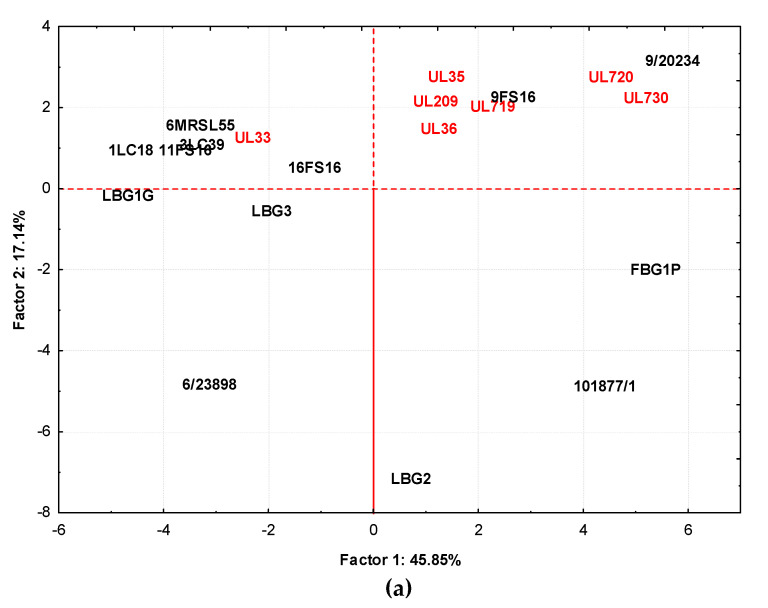
Distribution of cases (**a**) and variables (**b**) on the factor plane (1 × 2) obtained by samples fermented at 30 °C. Factor 1 and Factor 2 explain 45.85 and 17.14% of the total variance, respectively. The sample obtained by strains isolated from Canadian products are highlighted in red.

**Figure 8 foods-09-01815-f008:**
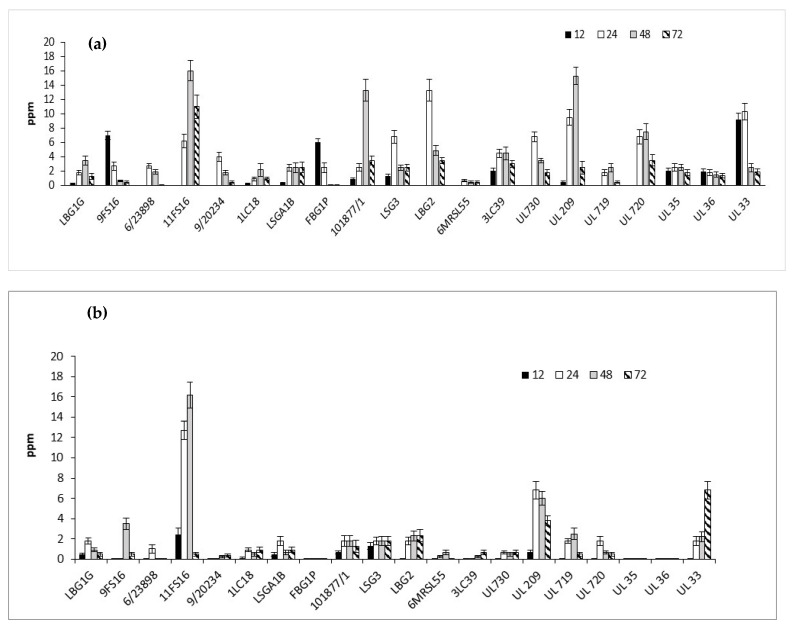
Detection of nisin (expressed as ppm) in milk samples fermented by Italian and Canadian strains at 30 (**a**) and 37 (**b**) °C.

**Table 1 foods-09-01815-t001:** *Lactococcus lactis* strains used in the research.

Strains	Species	Isolation Source	Produced Nisin	Collection
**Italian strains**				
3LC39	*Lactococcus lactis* subsp. *lactis*	Goat milk	A	UNIBO
LSGA1B	*Lactococcus lactis* subsp. *lactis*	Raw Cow milk	*	UNIBO
11FS16	*Lactococcus lactis* subsp. *lactis*	Fiore sardo cheese	A	UNICA
9/20234	*Lactococcus lactis* subsp. *lactis*	Sheep milk	A	UNICA
LSG3	*Lactococcus lactis* subsp. *lactis*	Raw Cow milk	*	UNIBO
10/18771	*Lactococcus lactis* subsp. *lactis*	Sheep milk	A	UNICA
LBG1G	*Lactococcus lactis* subsp. *lactis*	Raw Cow milk	*	UNIBO
FBG1P	*Lactococcus lactis* subsp. *lactis*	Raviggiolo cheese	*	UNIBO
LBG2	*Lactococcus lactis* subsp*. lactis*	Cow milk	*	UNIBO
9FS16	*Lactococcus lactis* subsp. *lactis*	Fiore Sardo cheese	A	UNICA
1LC18	*Lactococcus lactis* subsp. *lactis*	Goat milk	A	UNICA
6MRSL55	*Lactococcus lactis* subsp. *lactis*	Sheep’s milk	Z	UNICA
6/23898	*Lactococcus lactis* subsp. *lactis*	Sheep milk	A	UNICA
**Canadian strains**				
UL33	*Lactococcus lactis* subsp*. lactis* biovar *diacetylactis*	Cheddar cheese	A	STELA
UL35	*Lactococcus lactis* subsp*. lactis* biovar *diacetylactis*	Cheddar cheese	A	STELA
UL36	*Lactococcus lactis* subsp*. lactis* biovar *diacetylactis*	Cheddar cheese	A	STELA
UL209	*Lactococcus lactis* subsp*. lactis* biovar *diacetylactis*	Cheddar cheese	Z	STELA
UL719	*Lactococcus lactis* subsp*. lactis* biovar *diacetylactis*	Cheddar cheese	Z	STELA
UL720	*Lactococcus lactis* subsp*. lactis* biovar *diacetylactis*	Cheddar cheese	Z	STELA
UL730	*Lactococcus lactis* subsp*. lactis* biovar *diacetylactis*	Cheddar cheese	Z	STELA

* To be verified in the present research.

**Table 2 foods-09-01815-t002:** Values reporting the difference of cell density of *Listeria monocytogenes* Scott A grown without *Lactococcus lactis* inoculum and in the presence of *L. lactis* inoculum. *Listeria monocytogenes* Scott A was inoculated in milk at a level of 4.0 log cfu/mL in relation to the co-inoculated *L. lactis* strain (about 6.0 log cfu/mL) and the adopted temperature (30 and 37 °C). For each time considered, the difference was calculated with respect to the control sample represented by *L. monocytogenes* Scott A, inoculated in milk at a level of 4.0 log CFU/mL, and incubated at 30 and 37 °C. For each column, samples with a different lowercase letter are significantly different (*p* < 0.05).

Time (h)	12 h 37 °C	12 h 30 °C	24 h 37 °C	24 h 30 °C	48 h 37 °C	48 h 30 °C	72 h 37 °C	72 h 30 °C
101877/1	−0.2 ± 0.1 ^a^	−1.3 ± 0.2 ^b^	−1.9 ± 0.3 ^a^	−2.5 ± 0.3 ^b^	−4.0 ± 0.2 ^a^	−3.4 ± 0.2 ^b^	−5.7 ± 0.3 ^a^	−5.0 ± 0.2 ^b^
LSGA1B	−0.2 ± 0.2 ^a^	−0.4 ± 0.1 ^a^	−2.1 ± 0.2 ^a^	−2.7 ± 0.1 ^b^	−1.7 ± 0.2 ^a^	−1.7 ± 0.1 ^a^	−1.6 ± 0.1 ^a^	−0.9 ± 0.1 ^b^
6/MRSL55	−0.5 ± 0.2 ^a^	−0.3 ± 0.1 ^a^	−0.7 ± 0.1 ^a^	−1.3 ± 0.2 ^b^	−1.3 ± 0.1 ^a^	−0.8 ± 0.2 ^b^	−2.6 ± 0.2 ^a^	−1.9 ± 0.1 ^b^
LSG3	−0.6 ± 0.1 ^a^	−0.9 ± 0.1 ^b^	−0.7 ± 0.2 ^a^	−1.3 ± 0.1 ^b^	−1.4 ± 0.2 ^a^	−1.4 ± 0.2 ^a^	−3.0 ± 0.1 ^a^	−2.4 ± 0.3 ^b^
LBG2	−0.5 ± 0.1 ^a^	−0.4 ± 0.1 ^a^	−0.6 ± 0.1 ^a^	−1.2 ± 0.1 ^b^	−3.0 ± 0.4 ^a^	−2.9 ± 0.1 ^a^	−4.3 ± 0.2 ^a^	−3.9 ± 0.2 ^a^
9FS16	−0.9 ± 0.2 ^a^	−1.4 ± 0.3 ^b^	−2.0 ± 0.3 ^a^	−2.6 ± 0.1 ^b^	−2.6 ± 0.3 ^a^	−2.5 ± 0.2 ^a^	−4.2 ± 0.3 ^a^	−4.0 ± 0.1 ^a^
9/20234	−0.4 ± 0.1 ^a^	−0.5 ± 0.1 ^a^	−1.5 ± 0.1 ^a^	−2.5 ± 0.2 ^b^	−2.1 ± 0.4 ^a^	−2.1 ± 0.3 ^a^	−2.2 ± 0.1 ^a^	−1.0 ± 0.3 ^b^
LBG1G	−0.6 ± 0.3 ^a^	−0.4 ± 0.1 ^a^	−1.7 ± 0.4 ^a^	−2.4 ± 0.2 ^b^	−2.6 ± 0.3 ^a^	−2.3 ± 0.3 ^a^	−1.6 ± 0.5 ^a^	−1.3 ± 0.3 ^a^
FBG1P	−0.8 ± 0.1 ^a^	−1.8 ± 0.3 ^b^	−2.4 ± 0.1 ^a^	−3.7 ± 0.1 ^b^	−2.3 ± 0.1 ^a^	−4.1 ± 0.3 ^b^	−2.3 ± 0.3 ^a^	−5.0 ± 0.1 ^b^
3LC39	−0.8 ± 0.1 ^a^	−1.3 ± 0.1 ^b^	−2.8 ± 0.1 ^a^	−3.1 ± 0.2 ^b^	−2.3 ± 0.3 ^a^	−2.9 ± 0.3 ^a^	−4.3 ± 0.3 ^a^	−3.9 ± 0.2 ^a^
1LC18	−0.4 ± 0.2 ^a^	−0.4 ± 0.3 ^a^	−1.2 ± 0.1 ^a^	−2.0 ± 0.2 ^b^	−1.1 ± 0.1 ^a^	−1.2 ± 0.2 ^a^	−1.6 ± 0.2 ^a^	−0.9 ± 0.2 ^b^
11FS16	−0.5 ± 0.1 ^a^	−0.6 ± 0.2 ^a^	−1.1 ± 0.1 ^a^	−4.6 ± 0.3 ^b^	−2.3 ± 0.1 ^a^	−4.2 ± 0.3 ^b^	−3.3 ± 0.1 ^a^	−3.0 ± 0.1 ^a^
6/23898	−0.5 ± 0.2 ^a^	−0.3 ± 0.1 ^a^	−4.9 ± 0.3 ^a^	−5.1 ± 0.1 ^a^	−1.3 ± 0.2 ^a^	−1.2 ± 0.1 ^a^	−1.8 ± 0.1 ^a^	−1.0 ± 0.2 ^b^
UL33	−0.1 ± 0.1 ^a^	−1.2 ± 0.1 ^b^	−1.3 ± 0.4 ^a^	−2.1 ± 0.1 ^b^	−1.7 ± 0.3 ^a^	−2.0 ± 0.4 ^a^	−2.3 ± 0.5 ^a^	−2.9 ± 0.3 ^a^
UL35	−0.8 ± 0.2 ^a^	−1.2 ± 0.3 ^a^	−2.2 ± 0.1 ^a^	−2.9 ± 0.3 ^b^	−2.5 ± 0.1 ^a^	−3.1 ± 0.1 ^b^	−3.5 ± 0.3 ^a^	−3.3 ± 0.3 ^a^
UL36	−0.6 ± 0.2 ^a^	−1.1 ± 0.2 ^b^	−1.5 ± 0.1 ^a^	−3.1 ± 0.3 ^b^	−2.2 ± 0.1 ^a^	−3.4 ± 0.2 ^b^	−3.1 ± 0.2 ^a^	−3.9 ± 0.2 ^b^
UL209	−0.6 ± 0.1 ^a^	−0.5 ± 0.1 ^a^	−1.4 ± 0.3 ^a^	−2.8 ± 0.4 ^b^	−1.5 ± 0.4 ^a^	−3.0 ± 0.2 ^b^	−2.7 ± 0.1 ^a^	−3.1 ± 0.2 ^b^
UL719	−0.9 ± 0.1 ^a^	−0.7 ± 0.2 ^a^	−1.6 ± 0.3 ^a^	−2.8 ± 0.2 ^b^	−1.3 ± 0.3 ^a^	−3.0 ± 0.3 ^b^	−2.5 ± 0.3 ^a^	−2.9 ± 0.2 ^a^
UL720	−0.7 ± 0.2 ^a^	−0.8 ± 0.1 ^a^	−1.7 ± 0.3 ^a^	−2.8 ± 0.1 ^b^	−1.4 ± 0.4 ^a^	−2.9 ± 0.5 ^b^	−2.9 ± 0.2 ^a^	−3.7 ± 0.1 ^b^
UL730	−0.8 ± 0.3 ^a^	0.9 ± 0.1 ^a^	−1.7 ± 0.1 ^a^	−2.8 ± 0.3 ^b^	−1.6 ± 0.2 ^a^	−3.0 ± 0.2 ^b^	−2.6 ± 0.1 ^a^	−2.8 ± 0.3 ^a^

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
