# Peer review of "Unravelling the Potential of Lactococcus lactis Strains to Be Used in Cheesemaking Production as Biocontrol Agents"

_foods, 2020, doi:10.3390/foods9121815_

Round 1
Reviewer 1 Report
Dear authors
Most of the comments were considered, however, I still think the objective can be better. Here is one suggestion:
In this framework, the main objective of the present study, developed by an exchange program between Italy and Quebec, was to characterize selected L. lactis strains, isolated from Italian and Canadian dairy products in order to use some of them as potential protective cultures in cheesemaking to produce Cheddar (hard cheese) or Crescenza (soft cheese). Twenty strains from two geographical origins (Italy and Canada) were screened for (some) important technological traits and their ability to generate volatile molecule profiles in milk at two different incubation temperatures. Additionally, in order to select good biocontrol agents, the nisin production over the incubation and the ability to inhibit Listeria monocytogenes in skim milk were also screened for all the strains.
Line 190 Put in …. Figures 2 and 3, respectively.
Line 196 …. incubated at 30°C. (Table 3). Remove the first full stop. Where is Table 3?
Check table numbers!
L 332 aromagramma = aromagrams?
Author Response
Thank you very much to the Referee 1 for the suggestion. The text was modified according to and highlighted in red in the text.
Best Regards
Francesca Patrignani

Reviewer 2 Report
The authors proposed a revised and improved version of the publication “Unravelling the potential of Lactococcus lactis strains to be used in cheesemaking production as biocontrol agents”. They characterized different strains of L. lactis to select a consortium to be used as biocontrol agent. The subject is relevant for publication and the message is now clearer. Thanks to the authors for answering to my questions and for replacing tables by figures. However, some clarifications and corrections must be achieved to improve the final version.
More precisely
Table 1
The presentation is very clear now, but I have some remarks on the nomenclature. If you use the older assignation of species (by Schleifer et al 1985 or Rademaker et al 2007), please use “Lactococcus lactis subsp. lactis biovar diacetylactis” instead of “Lactococcus lactis biovar diacetylactis”. If you are not sure of the subsp, use Lactococcus lactis. However, if you prefer to use the revised assignation according to the most recent works (Torres Mano 2018, Cavanagh 2015), Lactococcus lactis group consists of the two species lactis and cremoris and biovar diacetylactis is related to L. lactis species. It is necessary to standardize the nomenclature. Please verify the italics for lactis.
Figure 1
Please, show only the sequences of amplicons you mentioned in the text (i.e. remove nis LSP2, nis FSP1 and the seq of nisA and Z, since you present the positive controls)
Minor remarks
- Line 102-104: “…a negative control and strains encoding for nisin Z (L. lactis CBM21, [22]) and nisin A (L. lactis ATCC11454, [23]) as positive controls were used”, instead of … a negative control, a strain encoding for nisin Z (L. lactis CBM21, [22]) and a strain encoding for nisin A (L. lactis 103 ATCC11454, [23]) as positive controls were used.
-Line 196: (Fig 4) instead of (Table3) and Fig 3 instead of Table 3
-Figure 2, 3, 4 and 5: Time 48h is not present in the legend. For Fig 3 and 5, I wonder why you don’t mentioned the statistical analysis as in Fig 2 and 4?
-Line 214: Add 6/23898 strain (the same significant decrease of pH as FBG1P and LBG2 at 30°C.
-Line 233: …”related to the biovar”, instead of …”related to the strain’s subspecies”
-Line 238: you mentioned ”LSGA1B” in the text , but it not on the Fig 6a
-Line 259: “11FS16” instead of “9FS16”
-Line 292: UL36 instead of Ul36
Author Response
Thank you very much to the Referee 2 for the suggestion. The text was modified according to and highlighted in red.
Best regards
Francesca Patrignani

Reviewer 3 Report
Requires minor corrections.
Ln 32-33 – what the authors mean by the sentence? Are strains generated in milk or volatile profiles?
Ln 44 Sometimes, cheese can be resistant to the pasteurization process of the raw materials. – Sentence has to be written more clearly.
Ln 55 spoiling -> spoilage
Ln 83 some -> certain?
Ln 186 – this is actually a multiple alignment of amplicon sequences…
Figures 2,3,4,5 are difficult to read – had to zoom to 200% to make it legible, legend missing 48 h pattern…do those lowercase letters match sampling times? It’s unclear from description…
Ln 225 & 332 – aromagramma? Couldn’t find this word in English…
Ln 269 skimmed
Author Response
Thank you very much to the Referee 3 for the suggestions. The text was modified according to and highlighted in red in the text.
best Regards
Francesca Patrignani

This manuscript is a resubmission of an earlier submission. The following is a list of the peer review reports and author responses from that submission.
Round 1
Reviewer 1 Report
The study by Bukvicki et al, considers the potential of different nisin producing Lactococcus lactis strains to be used as biocontrol agents in Crescenda and Cheddar cheese production. The strains have been isolated from these Italian and Canadian cheeses. First, the authors confirmed which form of nisin (A or its variant Z) is produced by sequencing nisin encoding gene. Then, the technological properties (growth kinetic, pH decrease, aromatic capacities) were evaluated. At last, nisin production and activity against Listeria monocytogenes were assessed.
In my opinion, this subject is of great interest in the field biopreservation. But, the conclusions are not always fully correct and/or must be clarified to report a relevant message.
More specifically:
Regarding detection of nisin encoding gene: the figure 1 is not of a good quality and could therefore be deleted. The authors mentioned a 598 bp fragment amplification in all strains considered. That is not the case for FSGA5A strains. This strain is not referenced in the table 1. Likewise with LSP2 and FSP1 not cited in Table1. The authors describe the Canadian strains as belonging to the subsp. diacetylactis, but it is nor a subdspecies but a biovar. Please clarify this section. There is no homogeneity when you mentioned the strains collections (collection of STELA (line 207), LAVAL collection (line 262)…
A minor comment: You precise that PCR were performed with 20 µg genomic DNA. It’s a very high quantity! Please confirm.
Regarding the growth kinetic, pH decrease and VOC: Tables 2, 3, 4 and 5 are difficult to read. Prefer histograms or curves. Likewise with Fig 4a.
Regarding nisin production: you mentioned that strain 9FS16 produce comparable amounts of nisin at 30°C and 37°C. It is not obvious in the Fig 5a. In my opinion that is the case for 11FS16 and UL209 but not for 9FS16. Please clarify these conclusions or modify the figure.
Regarding interactions with Listeria monocytogenes: in my opinion, some controls are needed: 1) growth of Listeria monocytogenes in presence of only nisin. Indeed, in the challenge test, many interactions are involved and the anti-listeria effect may be due to lactic acid produced by Lactococcus lactis or by other molecules or specific interactions between the two bacteria. 2) growth of Listeria monocytogenes in presence of a nisin deficient L. lactis. 3) It would be useful to test different Listeria monocytogenes strains.
The discussion is well written and quite interesting.
Author Response
We would like to thank the Referee 1 for the interesting suggestions able to improve the quality of the present manuscript. All the Referee suggestions were addressed according to and highlighted in red in the text.

Reviewer 2 Report
Response to the authors
In this manuscript the authors study different strains of L. lactis and their properties as biocontrol agents in dairy products. By producing nisin, the strains might control L. monocytogenes and contribute to taste and flavor of different cheese products.
The introduction covers the thematic in this study.
The objective was “to identify L. lactis on the basis of anti-listeria activity and nisin production. The strains had already been selected for their gene encoding nisin production.” (to characterize?)
In material and methods table 1 gives the information about the strains content of nisin gene A or Z. The following section on the work on detection of the nisin gene is therefore not necessary, just refer to the reference in the table. If this is work that is performed in this study, then the nisin gene content in table 1 must be removed and transferred to the results.
Line 107 could it be “positive control”?
Line 111 “50 ml” changed to “50 mL “ Throughout the manuscript there is an inconsistency in units of measurements. Check the journals author guidelines
The results are mainly presented as tables. Some of the results would be more readable if they were presented as e.g. graphs. Figure 1 The markers are not readable. There is definitely a gene there however can’t control the size.
Table 2 describes “growth kinetics” Growth kinetics is not what this table contains (CFU). The information in this table is about L. lactis is growing in milk, different speeds - different strains , calculate specific growth rate, m max.
Table 4 Growth kinetics? see comments table 2
Figur 5 The figure is difficult to read, the objective of this research is to select strains for further use. In this figure the colors of the columns are different for different sampling times.
Table 6 is difficult to interpret. As I understand it, it is log CFU/mL of Listeria monocytogenes Scott A (not scotta A) (check figure legends) co-inoculated with the different L. lactis strains. The start level of L. mono is 4.0 log CFU/mL. How is this calculated to get a reduction of -5? I think this would be a nice graph.
The discussion is OK, however some repeated sequences can be removed. (e.g. line 323 The strains….) L. lactis is mentioned in line 320.
An interesting question is are any of the strains able to inhibit L. mono and is that correlated with nisin production? Some answers are given in the text.
Ref (4) have been using L. lactis as a starter and presents the results differently but more readable.
This is a nice study, but needs a bit attention, text, language, tables and figures. A lot of work has been done in this field which must be reflected in this manuscript.
Author Response
We would like to thank the Referee 2 for the interesting suggestions able to improve the quality of the present manuscript. All the Referee suggestions were addressed according to and highlighted in red in the text.
